

# Variability of coastal downwelling circulation in response to high-resolution regional atmospheric forcing off the Pearl River Estuary

Wenfeng Lai[1], Jianping Gan[1,2]

[1]Department of Ocean Science and Department of Mathematics, Hong Kong University of Science and Technology, Hong Kong SAR, China
[2]Southern Marine Science and Engineering Guangdong Laboratory (Zhuhai), China

*Correspondence to*: Wenfeng Lai (laiwf@link.cuhk.edu.hk)

**Abstract.** We investigated the variabilities of coastal circulation and dynamics in response to spatiotemporally variable high-resolution atmospheric forcing off the Pearl River Estuary during the downwelling wind. Our investigation focused on the processes and dynamics of coastal downwelling circulation in response to variable atmospheric forcing of (1) single station observation, (2) global reanalysis data, and (3) a high-resolution regional atmospheric model. We found that the high-resolution atmospheric model significantly improved the representations of the near-surface wind and air temperature, and the ocean model driven by the high-resolution and spatially variable atmospheric forcing improved the circulation and associated hydrographic properties in the coastal ocean. Momentum and vorticity analyses further revealed that the cross-isobath water exchange was primarily governed by the along-isobath pressure gradient force (PGF), which was influenced by different components of the atmospheric forcing. The spatial-temporal variability of high-resolution wind forcing determined the strength and structure of coastal circulation, and improved estimates of cross-isobath transport and the associated PGF by refining the net stress curl and nonlinear advection of relative vorticity in the simulation. The high-resolution heat forcing can greatly improve the sea surface temperature simulation and adjust the nonlinear advection of relative vorticity, resulting in changes in cross-isobath transport.

## 1 Introduction

Accurate representations of momentum and heat fluxes are essential for accurate ocean modeling, especially for complex coastal zones where the forcing is highly influenced by land-sea interactions. In numerical ocean simulations, atmospheric forcing is generally obtained from an observation station within the study area (Zu and Gan, 2015), large-scale reanalysis data (Liu and Gan, 2020), or idealized wind assumptions (Xie and Eggleston, 1999). However, these common methods neither reflect nor resolve the complex regional variations of atmospheric fluxes in the coastal zone that result from local topographic features and land-air-sea interactions. Several factors, such as the representation of topography, land-atmospheric interactions, surface heat and moisture fluxes transports, and physical processes parameterizations, can influence the accuracy of meteorological parameter simulation, such as near-surface winds and air temperature (e.g., Cheng and Steenburgh, 2005; Singh et al., 2021). Inaccurate representation of wind fields can lead to incorrect ocean current



structures and significant biases in sea surface temperature (SST). Numerical ocean simulation can benefit from higher resolution atmospheric forcing which can help resolve, for example, oceanic mesoscale processes in ocean circulation (Jung et al., 2014).

We here explore the important factors which can influence the response of coastal ocean circulation and associated
dynamics to high-resolution atmospheric forcing in the Pearl River Estuary (PRE). We conduct comparative analyses of the simulated results forced with different resolutions of atmospheric fluxes. The PRE has a "trumpet" shape, with a width of 5 km at the northern end and 35 km wide at the southern end. Two longitudinally deep channels in the central and east regions connect the PRE to the adjacent shelf in the Northern South China Sea (NSCS) where isobaths are approximately parallel to the coastline, forming a strong cross-shelf topographic gradient. Local and highly spatially variable winds and strong land-
sea breezes have significant impacts on circulation patterns in the estuary-shelf area around PRE (Lai et al., 2021). Down-estuary winds can intensify the stratification of the water column and can also strengthen the along-estuary exchange circulation through wind-induced straining. In contrast, an up-estuary wind will enhance vertical mixing and weaken the estuarine circulation (Lai et al., 2018). During the summer/winter, the freshwater plume extends eastward/westward over the shelf, where it interacts with the monsoon-driven currents (Pan et al., 2020). Seaward gravitational convection characterizes
the circulation within the PRE, while the circulation in the lower estuary is governed by a combination of gravitational convection and intrusive geostrophic currents from the shelf (Zu et al., 2014). Throughout the NSCS, coastal downwelling and upwelling circulations are strongly influenced by the alongshore wind stress and wind stress curl (Gan et al., 2015; Lai and Gan, 2022). The coastal circulation and water properties in and near the PRE are sensitive to highly variable atmospheric forcing, such as wind and heat flux.

Higher-resolution regional atmosphere models can better reproduce the variability of atmospheric momentum and heat fluxes by resolving complex terrain and result in improved forecast skills (Agustsson and Olafsson, 2007). Akhtar et al. (2017) found that using a 9-km resolution atmosphere model resulted in improved simulations of wind speed and turbulent heat flux compared to using a 50-km resolution model. High-resolution regional atmospheric models provide significant new insights by parameterizing or explicitly simulating atmospheric processes over finer spatial scales. As an example, extreme
atmospheric events in the Mediterranean coastal region can be better simulated by resolving fine details of land use and topography and explicit convection simulations (Hohenegger et al., 2008).

Numerical modeling of estuarine and coastal systems is often limited by the quality of atmospheric forcing data used (Myksvoll et al., 2011). Upper-ocean currents and turbulent mixing are sensitive to small-scale variations in atmospheric forcing. According to Pullen et al. (2003), ocean current predictions were more accurate and skillful when using a 4-km
resolution atmospheric forcing over a 36-km spacing. Similarly, high-resolution atmospheric forcing allowed for more realistic reproduction of circulation in the Gulf of Lions due to better representations of the oceanic mixed-layer characteristics and dynamics in the ocean model (Langlais et al., 2009). Kourafalou and Tsiaras (2007) found that high-resolution atmospheric forcing allowed for a rapid oceanic response to large-scale wind variability in the northern Aegean Sea, resulting in reasonable changes in water properties and circulation. The precise wind stress and wind stress curl



structures near the coast from high-resolution forcing improved simulations of coastal jet in Eastern Boundary Upwelling Systems, leading to a significant reduction of warm biases in SST (Small et al., 2015). Finally, the capabilities to describe thermohaline circulation and water mass formation processes were improved under increased spatiotemporal resolution of atmospheric forcing at a regional scale (Castellari et al., 2000; Artale et al., 2016).

Accurate representation of atmospheric forcing is critical to correctly simulate the spatiotemporal variability of coastal
circulation and biogeochemical processes in coupled estuary-shelf systems. To thoroughly study the variability of the coastal ocean circulation, a combination of a state-of-the-art ocean model and high-resolution atmospheric forcing is required. In a recent study, Lai and Gan (2022) compared the impacts of low-resolution global reanalysis data and high-resolution wind forcing on the coastal circulation during upwelling-favorable winds and found that a high-resolution regional atmosphere model tended to produce stronger surface wind stresses, resulting in an intensified cross-isobath transport off the PRE.
However, despite this region is primarily dominated by the prevailing southwesterly monsoon (upwelling-favorable wind) during summer, the PRE also experiences episodic downwelling favorable wind, which can significantly alter the pathway of river plume, coastal circulation and cross-shore water transport. The alternations of the downwelling circulation under the spatiotemporal variable atmospheric fluxes and plume-induced buoyancy demonstrated the important wind-driven circulation dynamics.

In this study, our objective is to systematically investigate the responses of small-scale coastal circulation and associated dynamics to different resolution atmospheric forcings off the PRE during the downwelling-favorable wind. We aim to isolate the impacts of uniform wind, low-resolution wind, high-resolution wind, and heat flux forcing on the coastal downwelling circulation. To achieve this, we perform processes and dynamics analyses on the coastal wind-driven ocean circulation under forcing of momentum and heat fluxes using: (1) local station observations, (2) global reanalysis data, and (3) a high-
resolution regional atmospheric model. An improved understanding of the responses of oceanic processes and underlying dynamics to spatiotemporal variability of atmospheric forcing will advance our physical and numerical understanding and enhance the accuracy of coastal ocean modeling.

## 2 Models and observations

We run a high-resolution regional atmospheric model to dynamically downscale the large-scale reanalysis data and to obtain
fine atmospheric forcing for our coastal ocean model in the PRE. The surface heat budget and near-surface wind field derived from the model variables were used as surface boundary conditions for the ocean model.

### 2.1 The regional atmospheric model

We used the Weather Research and Forecast Model (WRF) to simulate the regional atmospheric environment in the PRE and NSCS. The WRF is a mesoscale, non-hydrostatic, numerical weather prediction model with advanced physics and
numerical schemes for simulating meteorology and dynamics (Michalakes et al., 1998). This model adopts a fully



compressible non-hydrostatic model with a horizontal Arakawa-C grid and terrain-following quality coordinates in the vertical direction.

A dynamical downscaling technique was employed in the WRF model to obtain a higher-resolution output (Caldwell et al., 2009) to better resolve the influence of regional dynamics and topography. A three-domain-nested configuration was
designed, as depicted in Figure 1a. The outer domain (D01) covered the western Pacific Ocean, the entire China Sea, and the Japanese Sea, and had a horizontal resolution of 9 km. The middle domain (D02) covered the NSCS with a horizontal resolution of 3 km. The inner domain (D03) had a high-resolution spacing of 1 km and covered the Guangdong-Hong Kong-Macao Greater Bay Area. The U.S. Geological Survey provided static geographical data, including land use and topographical information, with a horizontal resolution of 30 arc seconds (approximately 0.9 km). All domains had 33 sigma
layers in the vertical direction, and the maximum top-layer pressure was 50hPa. The initial conditions and lateral boundary conditions for the outer domain were provided by the Re-Analyses (ERA-interim) data from the European Centre for Medium-Range Weather Forecasts (ECMWF). The physics options of the WRF model included the WSM6 microphysics scheme (Hong et al., 2006a), CAM shortwave and longwave radiation schemes, YSU boundary layer scheme (Hong et al., 2006b), new Tiedtke cumulus scheme (Zhang and Wang, 2017) and Noah land surface model (Tewari et al., 2004). Results
from the D03 domain were used as the atmospheric forcing for the PRE coastal ocean model.

**2.2 Coastal ocean model and experiments**

The Regional Ocean Modeling System (ROMS) was used to simulate the coupled estuary-shelf hydrodynamic environment of the PRE (Liu and Gan, 2020). ROMS is a free-surface, hydrostatic, primitive equation model discretized with a terrain-following vertical coordinate system (Shchepetkin and McWilliams, 2005). The model domain covered the PRE and the
surrounding shelves in the NSCS with a grid matrix of 400×441 points (Figure 1b). The ultra-high-resolution grid (~0.1 km) resolved reasonably well the processes in the estuary and the inner shelf around Hong Kong. The grid size gradually increased to around 1 km over the shelf at the southern boundary of the domain. The model had 30 vertical levels with terrain-following coordinates (Song and Haidvogel, 1994) and adopted finer resolutions (<0.2 m) in the surface and bottom boundary layers to better resolve the dynamics there. The model was initialized and spun up with temperature and salinity
extracted from a well-validated larger-scale model covering the entire NSCS shelf (Gan et al., 2015). A new tidal and subtidal open boundary condition (TST-OBC) scheme developed by Liu and Gan (2020) was applied to this limited-area ocean model. Tidal forcing was applied to the open boundary using eight major tidal constituents (M2, K1, S2, O1, N2, P1, K2, and Q1) extracted from the Oregon State University Tidal Inversion Software (Egbert and Erofeeva, 2002).

Three experiments driven by different horizontal resolution wind forcings were implemented to assess the oceanic
response. The first experiment, referred to as WL-OBS, was forced by a temporally variable but spatially uniform wind field obtained from the Waglan Island meteorological station. The second experiment, referred to as LR-ERAI, was forced by the global reanalysis of ERA-interim data from ECMWF, including wind field and surface heat flux, with a spatial resolution of approximately 79 km. The third experiment, referred to as HR-WRFW, was forced by higher resolution (1 km) wind forcing



from the regional WRF model developed here. To isolate the effects of different wind forcing, we used the same surface heat
flux from the ERA-interim dataset, including temperature, pressure, solar radiation, and longwave radiation, in these three
experiments. We then conducted a fourth experiment forced by the high-resolution wind field and surface heat flux from the
regional WRF model, referred to as HR-WRFA, to investigate the impact of high-resolution heat flux. All ocean simulations
were initialized from the same initial condition and ran from 05 to 23 July 2017, covering a cruise period under a
downwelling-favorable wind.

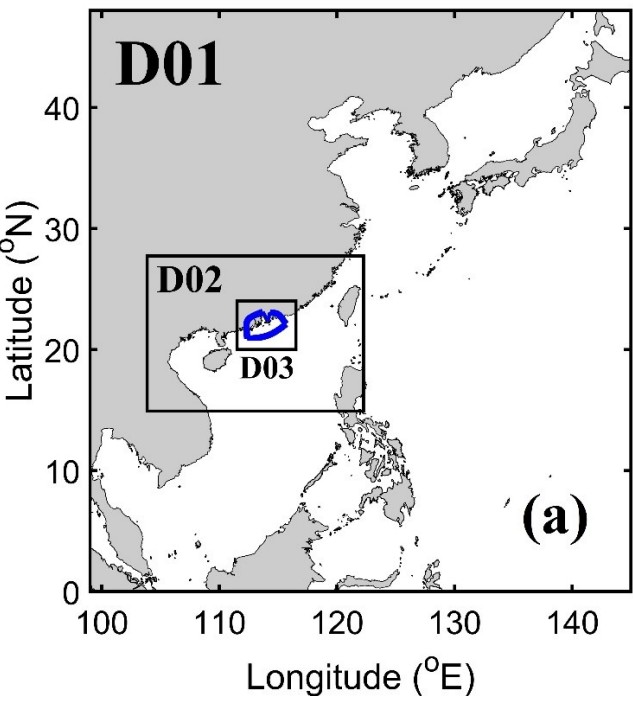


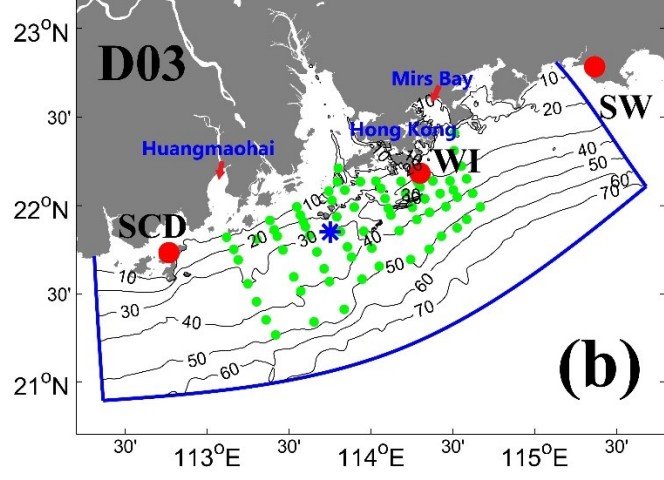



**Figure 1.** (a) The model domains of the WRF and PRE ocean models. The blue-lined area denotes the domain of the PRE ocean model. (b) The inner domain (D03) of the WRF model and the domain of the PRE ocean model with bathymetry (in units of m, black contour lines). The red dots indicate weather observation stations at Shan Wei (SW), Shang Chuan Dao (SCD), and Waglan Island (WI). The green dots denote the sampling stations of CTD during the cruise. The blue star indicates the location of the buoy.

## 2.3 Observations for validation

We used hourly surface wind and temperature data at weather stations around PRE obtained from the Hong Kong Observatory (HKO) and the Integrated Surface Database (ISD, https://www.ncdc.noaa.gov/isd), provided by the National Centers for Environmental Information (NCEI) and the National Oceanic and Atmospheric Administration (NOAA) to validate the atmospheric forcing.

We used conductivity-temperature-depth (CTD) data collected during a cruise survey from 13 to 21 July 2017 to validate the ocean model. In-situ salinity and temperature were measured using a well-calibrated Sea-Bird SBE25 CTD profiling system. The measurement stations were located along the transects in the PRE and over the adjacent shelf (Figure 1b). Profiles were extracted from the model results in the corresponding locations for validation against the CTD data. Simultaneously, we conducted time-series measurements of currents using a buoy mooring located south of Hong Kong Island (Figure 1b).

## 3 Model validations

### 3.1 Atmosphere model validation

We compared wind vectors and air temperature results from the atmospheric model to observations from the HKO and ISD. Figure 2 shows the along-shore and cross-shore winds and air temperature at the Shan Wei (SW), Waglan Island (WI), and Shang Chuan Dao (SCD) stations from the observations, the ERA-interim dataset, and the WRF model from July 11 to 23 2017. The alongshore and cross-shore components of the wind vectors are rotated by 23 degrees, with the alongshore component being approximately parallel, and the cross-shore component being approximately perpendicular to the coastline. A negative value of alongshore wind indicates a downwelling-favorable wind. Comparisons with observations at these three stations showed that the along-shore wind was the weakest at SW located to the east of PRE and strongest at WI outside the PRE (Figure 2). This suggests that the wind was highly variable along the coast and the WL-OBS experiment wind forcing was overestimated. This topographically-induced feature of the non-uniform spatial atmospheric forcing resulted in erroneously weaker variability of near-surface wind in the WL-OBS experiment.

At SW, the alongshore wind from the ECMWF (ERA-interim data) overestimated the downwelling-favorable wind on July 16 (Figure 2a), but it was underestimated at WI (Figure 2d). The low resolution of the ERA-interim dataset resulted in a similar magnitude and direction of the near-surface wind field at all stations. However, at WI, the intensities of the alongshore winds from the WRF model were stronger and closer to observations, indicating a better agreement between the



high-resolution modeled winds and observations. The along-shore winds at the SCD station were similar between the ERA-interim data and WRF, but the biases of cross-shore wind from the ERA-interim data were enlarged (Figure 2h). Overall, the

high-resolution WRF model improved the simulation of cross-shore winds and agreed better with the observations at all three observation stations (Figure 2b, e, h). The root-mean-square error (RMSE) of the alongshore and cross-shore winds from the WRF model were smaller than those from the ERA-interim data for the three stations (Table 1). In addition, the correlation coefficients of the alongshore and cross-shore winds for the WRF model were more significant.

The air temperature at WI dropped from nearly 30℃ on July 11 to ~ 25℃ on July 17 and then gradually increased to 30℃

again (Figure 2c). Both the ERA-interim data and WRF model results accurately reproduced this synoptic process, but the low-resolution ERA-interim data overestimated the air temperature at SCD (Figure 2i). The RMSE of the air temperature at SW was similar, but it was smaller at SCD for the high-resolution WRF result (Table 1). The correlation coefficient of air temperature was more significant in the high-resolution WRF model.

Overall, the reanalysis provided an unrealistic wind pattern in the coastal region, while a dynamical downscaling using a

high-resolution atmosphere model was necessary to accurately capture the observed spatial variability in atmospheric circulation. The high-resolution atmospheric model improved the representations of wind vectors and air temperature, with lower RMSE and higher correlation coefficients of wind and temperature, respectively.

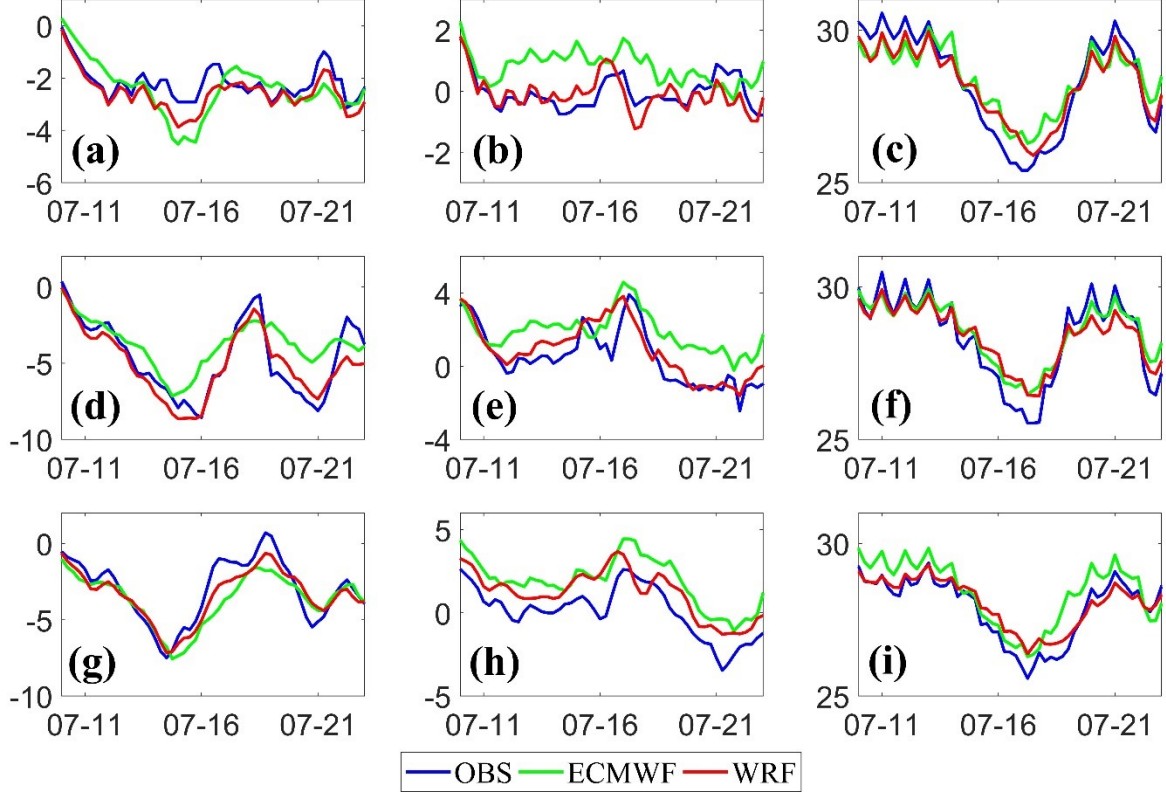





**Figure 2.** Comparisons of (a, d, g) along-shore and (b, e. h) cross-shore winds (ms$^{-1}$) and (c, f, i) air temperature (℃) of the observations, ERA-Interim data from ECMWF, WRF model at Shan Wei (a, b, c), Waglan Island (d, e, f) and Shang Chuan Dao (g, h, i) stations. The negative value of the along-shore wind indicates a downwelling favorable wind. A 24h high-frequency filter has been applied to the data.

**Table 1.** Correlation coefficients and RMSEs of along-shore and cross-shore winds and air temperature from the ERA-Interim data and WRF model at the atmospheric observation stations.

| | Station | | Shan Wei | Waglan Island | Shan Chuan Dao |
|---|---|---|---|---|---|
| Correlation Coefficient | Along-shore wind | WRF | 0.95 | 0.93 | 0.92 |
| | | ECMWF | 0.85 | 0.88 | 0.89 |
| | Cross-shore wind | WRF | 0.69 | 0.87 | 0.88 |
| | | ECMWF | 0.32 | 0.85 | 0.85 |
| | Air Temperature | WRF | 0.94 | 0.96 | 0.96 |
| | | ECMWF | 0.92 | 0.96 | 0.88 |
| RMSE | Along-shore wind (ms$^{-1}$) | WRF | 0.68 | 1.67 | 0.96 |
| | | ECMWF | 1.00 | 2.47 | 1.26 |
| | Cross-shore wind (ms$^{-1}$) | WRF | 1.13 | 1.52 | 1.26 |
| | | ECMWF | 1.57 | 1.79 | 1.82 |
| | Temperature (℃) | WRF | 0.65 | 0.73 | 0.53 |
| | | ECMWF | 0.66 | 0.64 | 0.73 |

## 3.2 Ocean model validation

We used the horizontal map of temperature and salinity RMSEs derived from the data in the sampling stations to assess the performance of the ocean model under different atmospheric forcings (Figure 3). The RMSEs of salinity and temperature of the water column at the CTD observation stations in the WL-OBS experiment are shown in Figures 3a and 3e, respectively.

The larger RMSEs of temperature primarily occurred near the mouth of the estuary and in the offshore region over the shelf in the WL-OBS experiment (Figure 3a). The differences in temperature RMSEs between the LR-ERAI and WL-OBS experiments showed that the ocean model forced by the ERA-interim data significantly reduced the RMSEs of temperature at most stations (Figure 3b), especially in offshore waters. indicating the importance of spatial variability in wind forcing. When the ocean model was driven by the high-resolution wind forcing (HR-WRFW experiment), the RMSEs of temperature were further reduced in the offshore region (Figure 3c). Along with the high-resolution heat flux forcing, the differences in



temperature RMSEs were negative at most stations (Figure 3d), particularly near the coast, showing that the minimum RMSEs of temperature can be obtained in the HR-WRFA experiment. However, the RMSEs and correlation coefficient of water temperature suggested that the overall improvements in the WL-OBS, LR-ERAI, and HR-WRFW experiments were limited due to the use of the same low-resolution surface heat flux forcing (Table 2). However, application of the concurrent high-resolution surface heat flux forcing and wind forcing reduced RMSEs of temperature, and the correlation coefficient of temperature was increased in the HR-WRFA experiment.

In the WL-OBS experiment, the RMSEs of salinity at the stations were larger in the western coastal waters, where dynamical variability was most intense due to the combined forcings of freshwater discharge, tides, and winds. The RMSEs of salinity were relatively small in the offshore waters (Figure 3e). The RMSEs of salinity increased outside the estuary but decreased on the western side when the ocean model was forced by the ERA-interim data (Figure 3f). Applying high-resolution wind forcing in the HR-WRFW experiment led to a decrease of the RMSEs of salinity in the shelf region (Figure 3g) due to better simulation of the wind-driven current. The salinity RMSEs were reduced at most stations in the HR-WRFA experiment (Figure 3h) as well. Overall, the RMSE and correlation coefficient of salinity were comparable in the LR-ERAI and WL-OBS experiments, with higher RMSEs and lower correlation coefficients. However, the RMSEs of salinity were significantly reduced in the HR-WRFW experiment and further decreased in the HR-WRFA experiment (Table 2). The correlation coefficient of salinity also increased when using high-resolution atmospheric forcing.

**Table 2.** RMSEs and correlation coefficients of water temperature and salinity in the CTD observations from the WL-OBS, LR-ERAI, HR-WRFW, and HR-WRFA experiments.

| Experiment | Water temperature (℃) | | Salinity (psu) | |
|---|---|---|---|---|
| | RMSE | Correlation Coefficient | RMSE | Correlation Coefficient |
| WL-OBS | 0.93 | 0.91 | 2.37 | 0.91 |
| LR-ERAI | 1.0 | 0.91 | 2.34 | 0.91 |
| HR-WRFW | 0.99 | 0.91 | 1.96 | 0.92 |
| HR-WRFA | 0.82 | 0.94 | 1.85 | 0.93 |









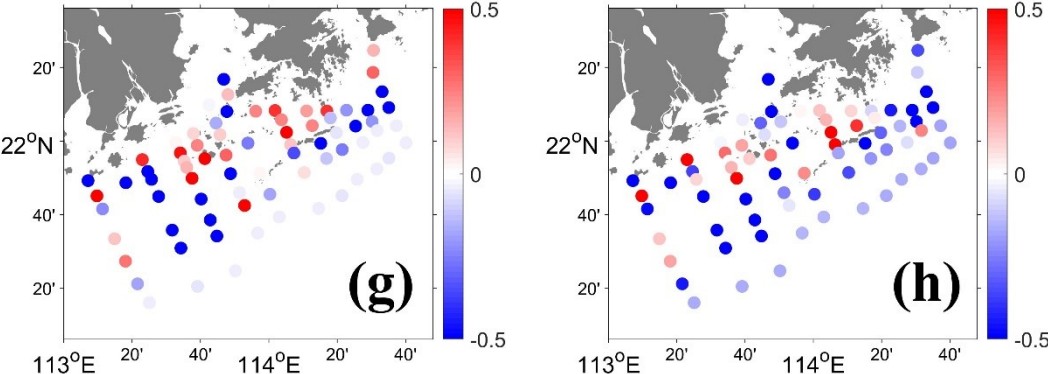

**Figure 3**. Horizontal map of RMSEs for (a) temperature (ºC) of the water column at the CTD stations in the WL-OBS experiment. The differences in RMSE of temperature between (b) the LR-ERAI and WL-OBS, (c) the HR-WRFW and LR-ERAI, and (d) the HR-WRFA and HR-WRFW experiments. Horizontal map of RMSEs for (e) the salinity (PSU) of the water column at the CTD stations in the WL-OBS experiment. The differences in RMSE of salinity between (f) the LR-ERAI and WL-OBS, (g) the HR-WRFW and LR-ERAI, and (h) the HR-WRFA and HR-WRFW experiments.

The time series of depth-averaged zonal and meridional velocities measured by the moored current meters were compared with the results from the different experiments (Figure 4). The recorded depth-averaged zonal and meridional velocities showed extensive fluctuations under a downwelling-favorable wind. Comparison with the results from the different experiments revealed that the biases of currents were larger in the WL-OBS and LR-ERAI experiments. However, using high-resolution atmospheric forcing decreased the biases of currents in the HR-WRFW and HR-WRFA experiments, resulting in better agreement with the observations. The correlation coefficient between the observed and simulated zonal velocity improved from 0.68 in the WL-OBS to 0.82 in the HR-WRFA experiment (Table 3). The RMSE of zonal velocity was also reduced from 0.091 ms$^{-1}$ to 0.063 ms$^{-1}$. The HR-WRFA experiment had the highest correlation coefficient and lowest RMSE for the meridional velocity.

In summary, the high-resolution atmospheric model more accurately captured the local dynamic effects induced by small-scale physical processes and terrain and simulated the wind better. The improved atmospheric forcing significantly improved the accuracy of simulated coastal currents, and consequently, the associated surface water temperature and salinity.





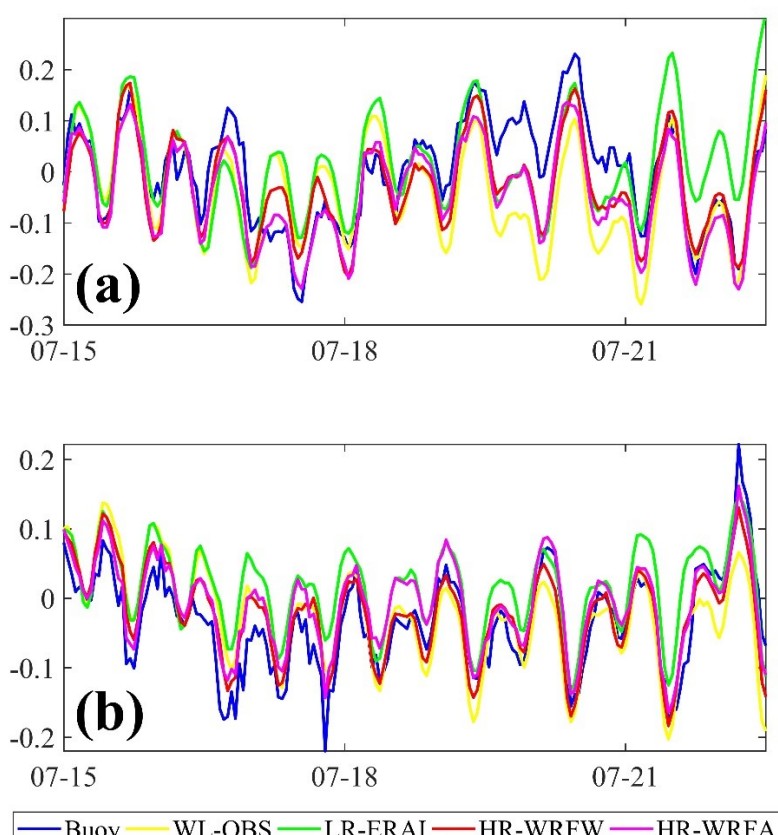

**Figure 4.** Time series of depth-averaged (a) zonal and (b) meridional velocities (ms$^{-1}$) at the buoy location from the buoy observations, the WL-OBS, LR-ERAI, HR-WRFW, and HR-WRFA experiments.

**Table 3.** RMSEs and correlation coefficients of depth-averaged zonal and meridional velocities (ms$^{-1}$) between buoy observations and the WL-OBS, LR-ERAI, HR-WRFW, and HR-WRFA experiments.

| | Zonal velocity | | Meridional velocity | |
|---|---|---|---|---|
| Experiment | RMSE | Correlation Coefficient | RMSE | Correlation Coefficient |
| WL-OBS | 0.091 | 0.68 | 0.056 | 0.75 |
| LR-ERAI | 0.086 | 0.62 | 0.063 | 0.83 |
| HR-WRFW | 0.089 | 0.72 | 0.057 | 0.78 |
| HR-WRFA | 0.063 | 0.82 | 0.042 | 0.86 |



## 4 Response to the downwelling-favorable wind

At the beginning of the field cruise in the PRE region, a moderate upwelling-favorable wind was observed from July 6 to 10, 2017. This was followed by a prolonged period of downwelling-favorable wind starting from July 11. In each of our four experiments here, our main focus was to study the response of coastal ocean circulation to the downwelling-favorable wind,

which occurred between July 11 and 21. 2017.

### 4.1 Simulated characteristics of shelf currents

The dynamics of coastal ocean circulation are greatly influenced by wind stress and wind stress curl, which play a crucial role in the upwelling and downwelling of water. The time-averaged wind stress and wind stress curl used in the WL-OBS, LR-ERAI, HR-WRFW, and HR-WRFA experiments are shown in Figure 5 when the downwelling-favorable wind prevailed

from July 11 to 21. The different resolutions of wind data revealed variations in intensity and patterns in the wind stress and wind stress curl structures over the PRE region. In the WL-OBS experiment, the wind forcing from the Waglan Island observation was uniformly applied throughout the region, as seen in Figure 5a. However, the intensity of the wind stress from the ERA-interim reanalysis data was found to be weaker than that in the WL-OBS experiment (Figure 5b). Additionally, the alongshore wind stress was stronger in the west and weaker in the east. Furthermore, wind stress

strengthened in the high-resolution WRF model (Figure 5c), particularly over the shelf. The wind stress curl was absent in the WL-OBS experiment (not shown). However, the ERA-interim data revealed a generally negative and spatially uniform wind stress curl over the shelf (Figure 5d). In contrast, the high-resolution WRF model displayed a marked spatial variability and structure in the wind stress curl (Figure 5e). The high-resolution WRF model depicted an intensification of wind stress curl near the coast, which exhibited a more detailed representation of wind stress curl variability. However, the magnitude of

wind stress curl was weaker in comparison to the low-resolution ERA-interim data (Figure 5e).







**Figure 5.** The time-averaged wind stress (Pa) from the (a) Waglan Island observation, (b) ERA-interim data, and (c) high-resolution WRF
model during the downwelling favorable wind from 11 to 21 July 2017. The colours show the magnitude of the alongshore wind stress.
The time-averaged wind stress curl ($Nm^{-3}$) from the (d) ERA-interim data and (e) WRF model during the downwelling favorable wind
from 11 to 21 July 2017.

When extensive and persistent north-easterly (downwelling-favorable) winds prevailed, the pre-existing northeast
upwelling shelf currents carrying buoyant waters rapidly reverted to a southwestward flow in the surface layer (Figure 6a).
However, in the east of PRE, the north-eastward shelf currents were weakened but maintained (Figure 6a), supported by the
river plume salinity (Figure 6e) and a cold-water belt from the upslope invasion of deep shelf water by the upwelling
circulation (Figure 7a). According to Liu and Gan (2020), an upslope invasion of the shelf waters in the east can be induced
by a counter-current established by the remote effect from exchange flows in the Taiwan Strait where the south-westerly
wind was persistent.

The difference in surface currents between the WL-OBS and LR-ERAI experiments suggested that the south-westward
along-shore jet was weaker, but the north-eastward jet was stronger in the LR-ERAI experiment due to the weaker wind
stress (Figure 6b). Furthermore, the south-westward alongshore and north-eastward jets were stronger in the HR-WRFW
experiment driven by high-resolution wind forcing than in the LR-ERAI experiment (Figure 6c). When combined with the
high-resolution heat forcing, water was generally warmer (Figure 7d) and the strength of the south-westward alongshore jet
was decreased in the HR-WRFA experiment, however, surface currents slightly increased over the shelf as compared with
the HR-WRFW experiment.

The surface salinity in the PRE is characterized by the offshore and eastward movement of the Pearl River plume during
pre-existing northeast upwelling. When the wind stress turns to downwelling-favorable winds in the WL-OBS experiment,
the freshwater from the estuary advected westward and offshore while lower salinity water was extended to the east by the
north-eastward jet on the eastern side (Figure 6e). The 27 PSU surface salinity contours indicated more freshwater was
constrained on the western side of the PRE due to the stronger south-westward alongshore jet in the WL-OBS experiment
(Figure 6e) as compared to other scenarios. The patterns of surface salinity from the HR-WRFW and HR-WRFA
experiments were similar to that in the WL-OBS experiment in the west, indicating similar currents. Meanwhile, freshwater
spread further eastward in the LR-ERAI experiment, suggesting the strongest eastward jet, but the eastward jets were weaker
when driven by the high-resolution wind forcing in HR-WRFW and HR-WRFA experiments.







**Figure 6.** The time-averaged surface currents (ms⁻¹) in the (a) WL-OBS experiment during the downwelling favorable wind from 11 to 21 July 2017. The differences in the magnitude of time-averaged surface current between (b) the WL-OBS and LR-ERAI, (c) the LR-ERAI and HR-WRFW, and (d) the HR-WRFW and HR-WRFA experiments during the downwelling favorable wind from 11 to 21 July 2017. The velocity vectors in (a)−(d) indicate the surface circulation in the WL-OBS, LR-ERAI, HR-WRFW, and HR-WRFA experiments. The





colour in (a) shows the magnitude of the surface circulation in the WL-OBS experiment. The colours in (b), (c), and (d) show the
difference in the magnitude of the surface circulation. The time-averaged sea surface salinity (PSU) in the (e) WL-OBS experiment during
the downwelling favorable wind and the contours of 27 PSU surface salinity in the WL-OBS, LR-ERAI, HR-WRFW, and HR-WRFA
experiments are shown in (e) as blue, green, yellow, and magenta contour lines, respectively.

**Figure 7.** The time-averaged SST (℃) in the (a) WL-OBS experiment during the downwelling favorable wind from 11 to 21 July 2017.
The differences in time-averaged SST between (b) the LR-ERAI and WL-OBS, (c) the HR-WRFW and LR-ERAI, and (d) the HR-WRFA
and HR-WRFW experiments during the downwelling favorable wind from 11 to 21 July 2017. The time-averaged net heat flux (Wm$^{-2}$) in
the (e) HR-WRFW and (f) HR-WRFA experiments during the downwelling favorable wind from 11 to 21 July 2017. A downward flux is
positive (heating).



The SST distribution showed a noticeable upwelling circulation, as evidenced by the cold-water belt to the east of the PRE under downwelling-favorable winds (Figure 7a). Despite the heat flux forcing being the same in the WL-OBS, LR-ERAI, and HR-WRFW experiments, the near-surface water was cooler in the LR-ERAI experiment (Figure 7b), showing the impacts of wind and current on the SST. The difference in SST between the LR-ERAI and HR-WRFW suggested that SST was slightly decreased when driven by the high-resolution wind forcing (Figure 7c) due to increased vertical mixing induced by strengthened wind stress. The positive net heat flux from the ERA-Interim data was smaller than that from the high-resolution WRF model (Figure 7e, 7f) due to the underestimation of net downward shortwave radiation in the low-resolution global re-analysis data (Figure not shown), resulting in an increased SST in the HR-WRFA experiment (Figure 7d). The warmer water can cause the ocean surface to expand and its density to decrease, which can cause the water to rise and affect the pressure gradient, leading to changes in ocean circulation.

## 4.2 Cross-isobath transport

We examined the cross-isobath components of depth-averaged bottom velocities over the 10m to 50m isobaths by projecting the simulated velocities onto the cross-isobath direction to illustrate the response of the cross-shelf exchanges to different atmospheric forcing during the downwelling-favorable wind (Figure 8). We defined the orientation of the isobath in the computational domain as the inverse tangent of the northward and eastward gradients of the topography. A positive (negative) value represented an onshore (offshore) transport of the shelf water perpendicular to the isobaths. Based on the pattern of the water exchange in different regions of our model domain, we selected two regions around the PRE for discussion (Figure 8a). In the west, we defined Region W to be on the shelf from the 30 m to 50 m isobaths. Region E was in the east from the 20 m to 50 m isobaths. However, due to the complexity of the complex topography and islands in the central region, we exclude a discussion of cross-isobath transport for that area.

The patterns of depth-averaged and bottom cross-isobath transport were qualitatively similar in the four experiments, and here only the results from the HR-WRFW experiments are shown (Figures 8a and 8b). Cross-shelf exchanges of the shelf and coastal waters mainly occurred over the irregular topography near the Huangmaohai Estuary in the west (Region W) and over a shallow trough off Mirs Bay in the east (Region E). These water exchanges were intensified at the locations where the isobaths meander, as previously reported by Liu et al. (2020). When a downwelling-favorable wind prevailed after an upwelling-favorable wind, the depth-averaged cross-isobath transports exhibited a net offshore transport for Region W and an onshore invasion on the eastern side for Region E (Figure 8a), resulting in a lower temperature around Hong Kong Island (Figure 7a). Meanwhile, the bottom cross-isobath transport displayed similar patterns to the depth-averaged cross-isobath transport (Figure 8b). The domain- and depth-averaged cross-isobath transport suggested that the depth-averaged cross-isobath transport in the LR-ERAI experiment was strongest for both onshore and offshore transport. The cross-isobath transport was slightly weaker in the HR-WRFA experiment driven by the high-resolution heat forcing, compared with the HR-WRFW experiment (Figure 8c). The depth-averaged cross-isobath offshore transport in Region W was weakest in the




WB-OBS experiment, but the onshore transport in Region E was similar to the HR-WRFW and HR-WRFA experiment (Figure 8c). Similarly, the bottom onshore and offshore transports were strongest in the LR-ERAI experiment, but the intensities of bottom onshore and offshore transport were slightly stronger in the HR-WRFA experiment, compared with the HR-WRFW experiment (Figure 8d). In the next section, we discuss the underlying dynamics of the changing transports in the different experiments.

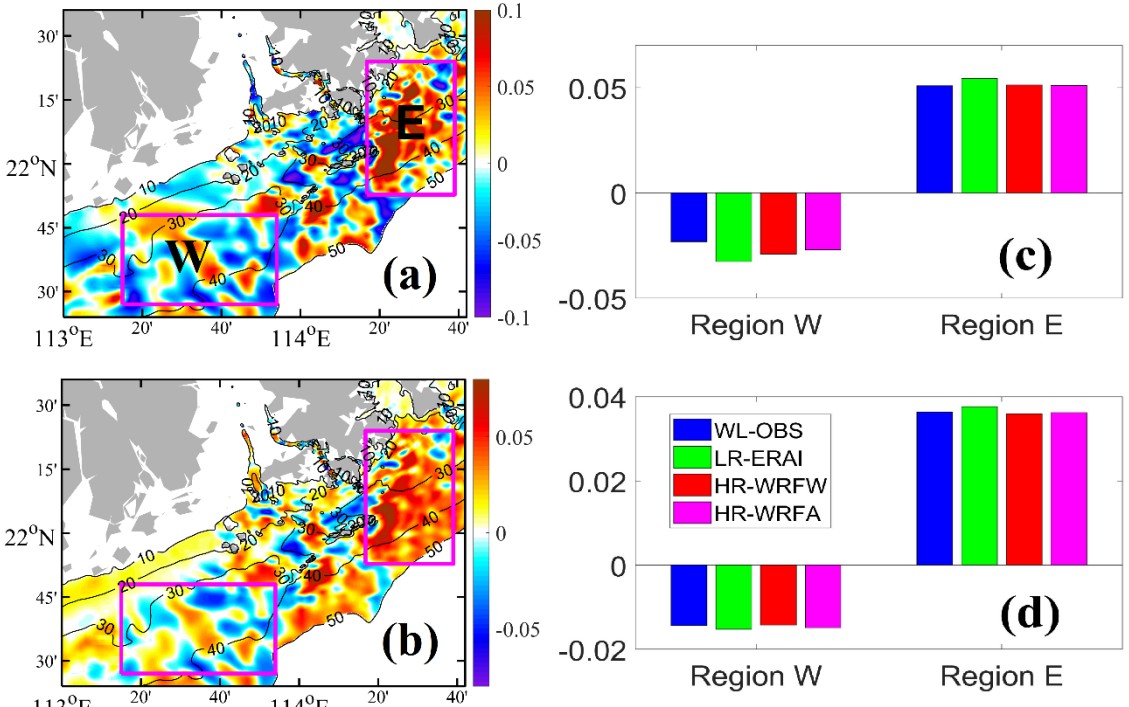

**Figure 8.** The time-averaged of (a) depth-averaged and (b) bottom cross isobath velocities (ms$^{-1}$) in the HR-WRFW experiment during the downwelling favorable wind from 11 to 21 July 2017. Time- and domain-averaged of the (c) depth-averaged and (d) bottom cross-isobath velocity (ms$^{-1}$) for Region W and E in the WL-OBS, LR-ERAI, HR-WRFW, and HR-WRFA experiments during the downwelling favorable wind from 11 to 21 July 2017. A positive (negative) value represents an onshore (offshore) transport of the shelf water

perpendicular to the isobaths.

## 5 Analyses of dynamics

To better understand the impact of high-resolution wind and heat forcings on the dynamic processes of the cross-isobath transport around the PRE, we use term balances in the depth-averaged momentum and vorticity equations. This enables us to examine the dynamic characteristics controlling the processes of cross-isobath transport.



## 5.1 Dynamics of cross-isobath transport

We analyzed the forcing mechanisms involved in the cross-isobath transport with the corresponding along-isobath momentum balances. The terms in the depth-averaged momentum equation were projected to the along-isobath direction to better investigate the cross-isobath velocity. The depth-averaged along-isobath momentum equation is expressed as

$$\overbrace{\frac{\partial \bar{v}}{\partial t}}^{ACCEL_{y*}} = \overbrace{-f\bar{u}}^{COR_{y*}} \overbrace{-[(\bar{u},\bar{v}) \cdot \nabla]\bar{v}}^{HADV_{y*}} \overbrace{-\frac{P_{y*}}{\rho_0 D}}^{PGF_{y*}} + \overbrace{\frac{\tau_{sy*}}{\rho_0 D}}^{SSTR_{y*}} \overbrace{-\frac{\tau_{by*}}{\rho_0 D}}^{BSTR_{y*}} + \overbrace{K_h \nabla^2 \bar{v}}^{HVISC_{y*}} \quad (1)$$

where the subscript, $y*$, denotes the momentum in the along-isobath direction. The $\bar{u}, \bar{v}$ are the depth-averaged velocities in the cross-isobath and along-isobath directions. The reference density and coefficient of horizontal viscosity are represented by $\rho_0$ and $K_h$, respectively. The terms in Eq. (1) are acceleration ($ACCEL_{y*}$), Coriolis force ($COR_{y*}$), horizontal nonlinear advection ($HADV_{y*}$), pressure gradient force ($PGF_{y*}$), surface stress ($SSTR_{y*}$), frictional bottom stress ($BSTR_{y*}$), and horizontal viscosity ($HVISC_{y*}$).

We first present time-averaged horizontal distributions of depth-averaged along-isobath momentum terms in the HR-WRFW experiment (Figure 9) during the downwelling-favorable wind. The corresponding differences in different experiments are given in Figure 10. The negligible contributions from the acceleration and $HVISC_{y*}$ were excluded from the following discussion. The principal balance along the isobaths in both Region W and E was between the $COR_{y*}$ and $PGF_{y*}$ terms (Figure 9a, 9b). The contributions of the $SSTR_{y*}$, $BSTR_{y*}$, and $HADV_{y*}$ were weaker than that of the $PGF_{y*}$, indicating that the major contributor to the cross-isobath exchanges was the along-isobath pressure gradient $PGF_{y*}$, suggesting a geostrophic current. A positive $PGF_{y*}$ in the west indicated an offshore transport ($COR_{y*}<0$), while a negative $PGF_{y*}$ in the east indicated an onshore transport ($COR_{y*}>0$). A negative $SSTR_{y*}$ and $BSTR_{y*}$ over the shelf suggests that surface and bottom Ekman processes contributed to the onshore invasion of shelf waters, weakening the offshore transport in the west and strengthening the onshore transport in the east. The $SSTR_{y*}$ and $BSTR_{y*}$ in Region W were weaker than those in Region E (Figure 9c). The $HADV_{y*}$ was weaker over the shelf but stronger in the navigational channels and around the islands, where the $PGF_{y*}$ was mainly balanced by $HADV_{y*}$ (Figure 9b, 9e), suggesting gravitational circulation.





**Figure 9.** Horizontal maps of time-averaged along-isobath (a) COR, (b) PGF, (c) SSTR, (d)BSTR, and (e) HADV (ms$^{-2}$) in the HR-WRFW experiment during the downwelling-favorable wind from 11 to 21 July 2017.

Figure 10 shows the time- and domain-averaged of the depth-averaged along-isobath momentum terms in the regions W and E for the WL-OBS, LR-ERAI, HR-WRFW, and HR-WRFA experiments during the downwelling-favorable wind. Similarly, the major balance along the isobaths in both Region W and E was between the $COR_{y*}$ and $PGF_{y*}$ in all four experiments. The intensity of the along-isobath $PGF_{y*}$ plays a significant role in determining the strength of cross-isobath transport in both Region W and E. Stronger along-isobath $PGF_{y*}$ in the LR-ERAI experiment resulted in stronger cross-isobath transport in both Region W and E. However, when high-resolution wind forcing was applied in the HR-WRFW





experiment, the intensities of the along-isobath $PGF_{y*}$ decreased in both regions W and E. When using the high-resolution

heat forcing in the HR-WRFA experiment, the intensities of the $PGF_{y*}$ were further reduced. The impact of $SSTR_{y*}$ on offshore transport in Region W varied in different experiments. In the WL-OBS experiment, using a spatially uniform wind forcing ($SSTR_{y*}$) weakened the offshore transport. In contrast, $SSTR_{y*}$ intensified the offshore transport in Region W in the LR-ERAI, HR-WRFW, and HR-WRFA experiments, which were driven by spatially variable winds. In Region W, the effect of $SSTR_{y*}$ on offshore transport in the LR-ERAI experiment was weaker than those in the HR-WRFW and HR-WRFA

experiments. However, in Region E, the effect of $SSTR_{y*}$ on the onshore transport was stronger in the HR-WRFW and HR-WRFA experiments due to enhanced wind stress, compared with the LR-ERAI experiment. $HADV_{y*}$ had a minimal impact on the cross-isobath transport in both Region W and E across all four experiments. $BSTR_{y*}$ had the strongest influence in the LR-ERAI experiment due to stronger bottom currents. Overall, the cross-isobath transport was primarily determined by the along-isobath $PGF_{y*}$ and adjusted by the $SSTR_{y*}$ and $BSTR_{y*}$ induced by the variable atmospheric forcing.

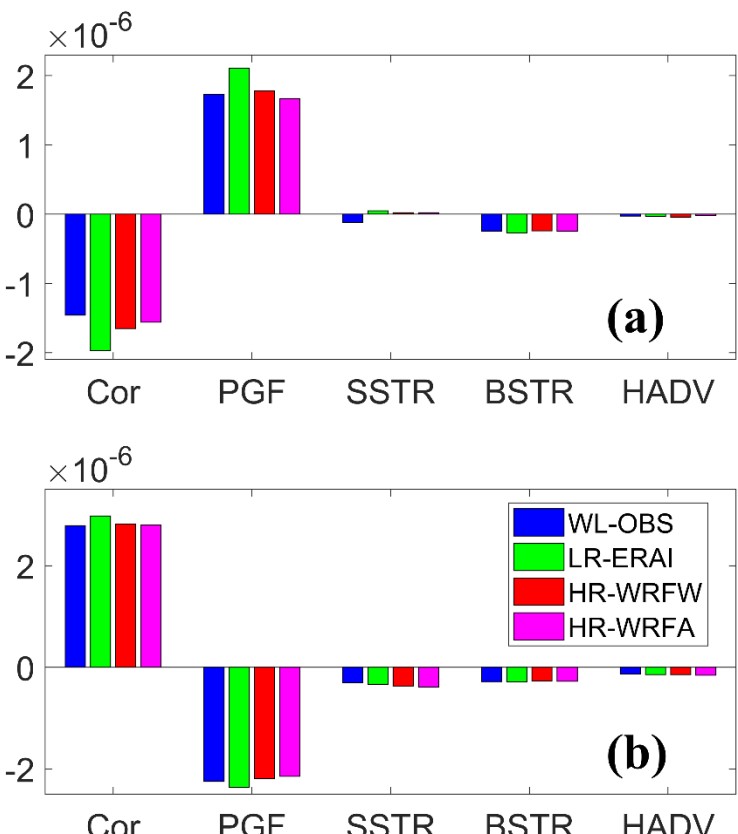


Figure 10. Time- and domain-averaged of along-isobath COR, PGF, SSTR, BSTR, and HADV (ms$^{-2}$) in Equation (1) for the WL-OBS, LR-ERAI, HR-WRFW, and HR-WRFA experiments in (a) Region W, and (b) E during the downwelling favorable wind from 11 to 21 July 2017.



## 5.2 Dynamics of along-isobath PGF

We identified that the cross-isobath exchange in the study area is primarily controlled by the geostrophic current from the $PGF_{y*}$. The cross-isobath velocity also showed great similarity to that in the bottom boundary layer in accordance with the small Ekman number. The impact of different atmospheric forcings on the formation mechanism of this along-isobath PGF ($PGF_{y*}$) is further explained in the following sections, utilizing the depth-integrated vorticity equation proposed in Gan et al. (2013) and Liu et al. (2020) (Eq. (2)):

$$\overbrace{\frac{-1}{\rho_0}P_{y*}}^{PGF_{y*}} = \overbrace{\frac{1}{D_{x*}}\nabla \times \left(-\frac{\frac{g}{\rho_0}\int_{-H}^{0}z\rho dz\nabla H}{H}\right)}^{JEBAR} + \overbrace{\frac{1}{D_{x*}}\bar{\nabla} \times \left(\frac{\tau_b}{\rho_0}\right)}^{BSC} + \overbrace{\frac{1}{D_{x*}}\bar{\nabla} \times \left(-\frac{\tau_s}{\rho_0}\right)}^{SSC} + \overbrace{\frac{1}{D_{x*}}J(\psi, \nabla \times \vec{v})}^{RVA} + \overbrace{\left(\frac{\|\vec{v}\|^2}{2}\right)_{y*}}^{GMF} \qquad (2)$$

where subscripts, $x*$ and $y*$, denote partial differentiation in the perpendicular and along isobaths directions, respectively. In the water column, the along-isobath $PGF_{y*}$ can be further decomposed into the JEBAR and pressure gradient force in the bottom boundary layer $PGF_{y*}^b$ (Mertz and Wright, 1992), as shown in Eq. (3):

$$PGF_{y*} = JEBAR + PGF_{y*}^b \qquad (3)$$

$PGF_{y*}$ is governed by the joint effect of baroclinic and relief (*JEBAR*), the net stress curl jointly governed by the bottom stress curl (*BSC*) and the surface stress curl (*SSC*), and the nonlinear relative vorticity advection (*RVA*), as well as the gradient of momentum flux (*GMF*). *GMF* can be neglected here due to its limited contribution compared with the other processes.

Figure 11 displays the time-averaged horizontal distributions of the bottom along-isobath PGF ($PGF_{y*}^b$), *JEBAR*, *BSC*,
*SSC*, and *RVA* terms in the HR-WRFW experiment during the downwelling-favorable wind. The spatial distribution of $PGF_{y*}^b$ was similar to the $PGF_{y*}$ (Figure 9a), which controlled the cross-isobath transports of shelf waters. *JEBAR*, caused by the baroclinicity of the buoyant plume and shelf waters, dominated the cross-isobath velocities near the coast and two deep channels (Figure 11b). *JEBAR* strengthened the onshore transport in Region E but weakened the offshore transport in Region W. *RVA*, due to the combined effect of the shelf current and the along-isobath variation of the relative vorticity, was the
major contributor to the water transport over the shelf (Figure 11e). *BSC* distribution was similar to that of *RVA* (Figure 11c), as the second-largest contributor to the cross-isobath exchanges of waters. *SSC* was smaller than other terms and systematically facilitated the cross-isobath transport of shelf waters in the water column (Figure 11d). *RVA*, *BSC*, and *JEBAR* regulated the cross-isobath exchanges of waters in Region W. In Region E, it was mainly sourced from *RVA*, and partially offset by *BSC* over the trough near the entrance of Mirs Bay.






**Figure 11.** Horizontal maps of time-averaged (a) bottom along-isobath PGF, (b) JEBAR, (c) BSC, (d) SSC, and (e) RVA (ms$^{-2}$) in the HR-WRFW experiment during the downwelling favorable wind from 11 to 21 July 2017. A positive (negative) value represents an onshore
(offshore) transport of the shelf water perpendicular to the isobaths.

The domain average of the along-isobath $PGF_{y*}$, *JEBAR*, *BSC*, *SSC*, and *RVA* terms in the regions W and E for the WL-OBS, LR-ERAI, HR-WRFW, and HR-WRFA experiments are shown in Figure 12 to further identify the impacts of different atmosphere forcing on the different components of forming the along-isobath PGF. A positive (negative) value indicates an
onshore (offshore) transport of the shelf water perpendicular to the isobaths. The net stress curl (*BSC* and *SSC*) and *RVA* contribute to the offshore transport in Region W for all the experiments (Figure 12a). In the HR-WRFA experiment driven



by high-resolution wind and heat forcing, *RVA* was strongest, but it was offset by a large *JEBAR*, resulting in a weaker along-isobath PGF and offshore transport. The LR-ERAI experiment had the weakest *BSC* and *RVA*, but the strongest *SSC*, leading to the strongest along-isobath PGF in Region W. The high-resolution WRF model tended to weaken the domain-averaged *SSC* but enhanced *BSC*, *RVA,* and *JEBAR*. In comparison to the LR-ERAI experiment, the HR-WRFW experiment saw an increase in *JEBAR*, which was further enhanced in the HR-WRFA experiment. In Region E, *RVA* was the major contributor in four experiments (Figure 12b). The LR-ERAI experiment had the strongest *RVA*, leading to a strong along-isobath PGF and cross-isobath transport. Similarly, *SSC* decreased and *BSC* increased in the HR-WRFW and HR-WRFA experiments driven by high-resolution wind forcing. The HR-WRFA experiment, which involved stronger net heat flux forcing, resulted in a weaker *RVA*. This was further offset by a larger *BSC*, leading to a weaker along-isobath PGF and cross-isobath transport.

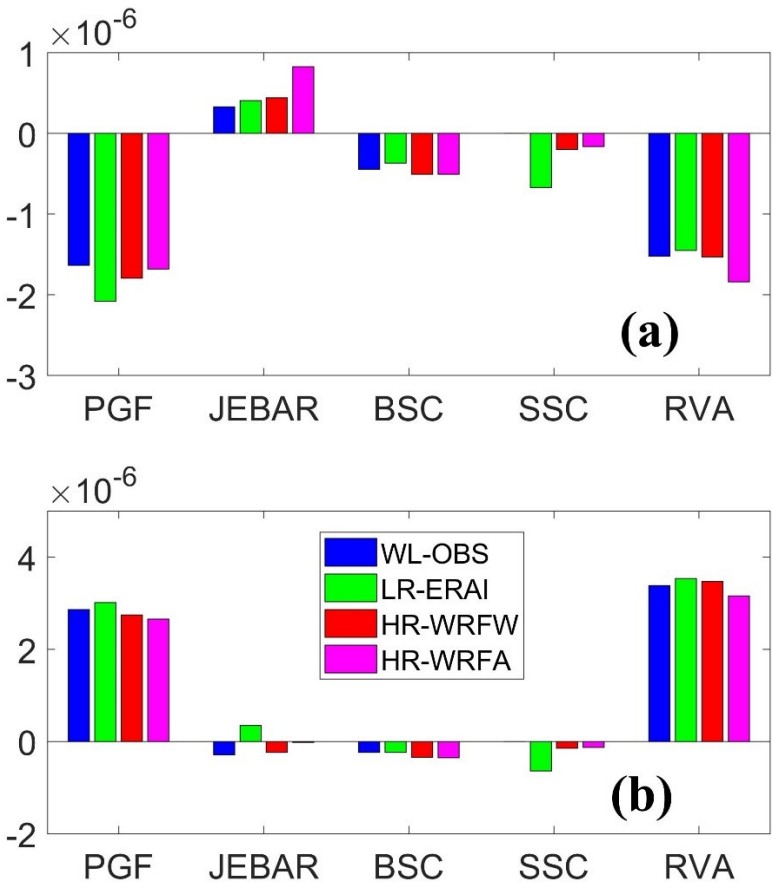

**Figure 12.** Time- and domain-averaged of along-isobath PGF, JEBAR, BSC, SSC, and RVA (ms$^{-2}$) in Equation (2) for the WL-OBS, LR-ERAI, HR-WRFW, and HR-WRFA experiments for (a) Region W, and (b) E during the downwelling favorable wind from 11 to 21 July 2017. A positive (negative) value represents an onshore (offshore) transport of the shelf water perpendicular to the isobaths.



Overall, the high-resolution WRF model weakened *SSC* and strengthened *BSC. BSC* and *SSC* are similar in the HR-WRFW and HR-WRFA experiments due to the similar wind forcing. The along-isobath PGF was influenced by the high-resolution heat forcing primarily through *JEBAR* and *RVA* processes, with *JEBAR* being stronger in the HR-WRFA

experiment. In the HR-WRFA experiment, *RVA* increased in Region W with a downwelling circulation but decreased in Region E with an upwelling circulation.

## 6 Summary

Accurate representation of the surface heat budget and momentum fluxes is essential to the boundary forcing of ocean modeling. We here investigated the variability of coastal circulation and dynamic response to spatiotemporal varying

atmospheric forcing off PRE at the onset of downwelling-favorable wind after a prevailing upwelling wind. Four numerical ocean simulations were driven by (1) the observations from Waglan Island station using a uniform spatial distribution, (2) global reanalysis data from ECMWF, (3) high-resolution wind forcing, and (4) high-resolution atmospheric wind and heat forcing from a regional atmospheric model. Comparisons between the model results and the in-situ observations suggest that the high-resolution atmospheric model significantly improved the simulation of the near-surface wind and air temperature,

resulting in significant accuracy improvements in our simulations for coastal ocean currents, water temperature, and salinity.

We also performed momentum and vorticity analyses to further rationalize the response of the cross-shore exchange flows to different atmospheric forcing factors. The momentum analyses suggest that the cross-isobath exchange of water was mainly governed by the along-isobath pressure gradient force, whose magnitude was affected by the different atmospheric forcings. Compared to the simulation driven by the lower resolution forcing, the intensity of along-isobath PGF decreased by

applying the high-resolution wind and heat flux forcings, resulting in a weaker cross-isobath water exchange. The frictional bottom and surface Ekman transport play a secondary role in altering the cross-isobath exchange. The high-resolution atmosphere forcing tended to strengthen the surface Ekman transport in the east of the PRE but weakened in the west.

Analyses based on the depth-integrated vorticity equations further suggest that the variability of the along-isobath PGF was jointly formed by the nonlinear advection of relative vorticity caused by the dynamic interaction between the eastward

flow and variable shelf topography and the net stress curl (frictional bottom and surface stress curl). The JEBAR effect efficiently adjusted the along-isobath PGF in the west but had limited impact in the east. The horizontal distribution of these terms was regulated by different atmospheric forcing to form the exchange flows. Stronger heat flux forcing increased the nonlinear advection of relative vorticity and the JEBAR effect but decreased the surface stress curl, resulting in a decreased along-isobath PGF and weakened cross-isobath water exchange in the west of the PRE. However, in the east, the nonlinear

advection of relative vorticity decreased, leading to a weaker cross-isobath transport. Improved high-resolution atmospheric forcing dynamically refined the formation and variability of the along-isobath pressure gradient force and the associated cross-isobath water exchange.



**Competing interests**

The authors declare that they have no conflict of interest.

**Data availability**

Model and observational data is available on request.

**Author contribution**

All authors contributed to the study conception and design. Material preparation, data collection, and analysis were performed by WF.L. The draft of the paper was written by WF.L and all authors commented on previous versions of the paper. All authors read and approved the final paper.

**Acknowledgments**

This work was supported by the Innovation Group Project of Southern Marine Science and Engineering Guangdong Laboratory (Zhuhai) (projects 311020003 and 311021004), the Theme-based Research Scheme (T21-602/16-R, GRF16212720) of the Hong Kong Research Grants Council. We are grateful for the support of The National Supercomputing Centers of Tianjin and Guangzhou. The numerical results presented in the figures were produced by solving
equations in the Regional Ocean Modeling System (ROMS). The source code of ROMS can be downloaded from https://www.myroms.org/.

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
