# Peer review of "Variability of coastal downwelling circulation in response to highresolution regional atmospheric forcing off the Pearl River Estuary"

_EGUsphere, 2023_

## Referee Comment (RC1)

**Review of "Variability of coastal downwelling circulation in response to high-resolution regional atmospheric forcing off the Pearl River Estuary"**

This paper by Lai and Gan investigated the variabilities of coastal circulation and dynamics in response to spatiotemporally variable high-resolution atmospheric forcing off the Pearl River Estuary during the downwelling wind. The authors conducted three numerical experiments based on (1) single station observation, (2) global reanalysis data and (3) high-resolution regional atmospheric model. Results show that the model with high-resolution atmospheric forcing significantly improved the temperature-salinity profiles and ocean current simulation. In addition, the model with high atmo-forcing improved the estimation of cross-isobath transport. This paper is well-written and organized, and the results are beneficial for ocean modeler to improve their model results and associated studies. I would suggest a minor revision for this paper before publishing it.

Major Comments:

(1) One of the comment problems in ocean model is the over-heating in the surface layer. Usually the ocean model needs surface temperature nudging to the reanalysis or climatology SST data, such as GHRSST and MODIS, to avoid over-heating in the surface layer. In this paper, the author mentioned that ROMS were forced by high-resolution wind stress and heat fluxes. I wonder whether ROMS model in this paper only driven by high-resolution heat fluxes without SST nudging, or only driven by high-resolution SST. Please clarify it in the discussion.

(2) The author used the ERA data with 75 km resolution as the "coarse" resolution product to compared with the WRF 1 km production. Actually, in nowadays, the 12-15 km resolution and 0.2 degree (approximately 22 km) resolution products are quite common, and provided by ECWMF and CFSV2 (from NECP), respectively. Ocean model driven by this 10-20 km resolution products may be closed to that

driven by WRF 1 km product. The author may give some comment on this.

---

## Author Comment (AC2)

*Lai and Gan 2023 analyse the variability of coastal circulation and dynamics in response to different atmospheric forcing during a period of downwelling-favorable winds. The analysis is similar to the one by the same authors (i.e. Lai and Gan 2022, cited) for upwelling winds (10-28th July 2015), while in this case they analyse a period of downwelling-favorable winds (5-23 July 2017). They analyse the sensitivity of the results to different spatio-temporal variable atmospheric forcing, namely 1) single station observation (WL-OBS), 2) global reanalysis data (LR-ERAI) and 3) high resolution regional atmospheric forcing (HR-WRFW with heatflux from ERAI and WRFA with heatflux from WRF) while in Lai and Gan 2022 they used 1) global reanalysis data, 2) high resolution regional atmospheric forcing and 3) air-sea coupled model. The results show that the model with high resolution forcing (and hence better representation of near-surface wind and air temperature) improved the simulation of coastal ocean currents, water temperature and salinity, and estimates of the across-isobath transport. The paper is well written and logically organized. The results are probably of interest for the community. I would suggest publication after minor revison.*

**Response:** Thank you and we appreciate the reviewer's time and comment.

*Minor comments*
*It is not clear from the text whether heat flux is used in the simulations forced with observational data. In The methods it only mentions the wind, however on L324 it states "…the heat flux forcing being the same in the WL-OBS, LR-ERAI, and HR-WRFW experiments…"*
**Response:** We used the same surface heat flux from the ERA-interim dataset to drive the ocean model in these three experiments. We clarify this in section 2.2 of the revised ms..

"The first experiment, referred to as WL-OBS, was forced by a temporally variable but spatially uniform wind field obtained from the Waglan Island meteorological station. The second experiment, referred to as LR-ERAI, was forced by the temporal-spatial variable wind forcing form the global reanalysis of ERA-interim data provided by the ECMWF, with a spatial resolution of approximately 79 km. The third experiment, referred to as HR-WRFW, was forced by higher resolution (1 km) wind forcing from the regional WRF model developed here. To isolate the effects of different wind forcing, we used the same surface heat flux based on bulk formula from the ERA-interim dataset to drive the ocean model in these three experiments, including temperature, pressure, solar radiation, and longwave radiation."

*The authors used ERAI that has ~75km resolution and 6h resolution as forcing for the LR-ERAI case and as boundary and initial conditions for the production of the WRF 1km that is then used as forcing for the high-resolution cases. ERA5 is available at ECMWF for the period of interest and has higher temporal and spatial resolution (~30km and hourly resolution). A model forced with this higher resolution dataset (ERA5) may provide more accurate results.*

**Response:** Yes, we acknowledge that latest ERA5 data with a 0.25-degree resolution has a higher temporal and spatial resolution compared to ERA-Interim data. In this study, we also employed the latest ECMWF ERA5 data to force the ocean model and obtained results that were comparable to those driven by the ERA-Interim data. The figure below shows the comparisons of the along-shore and cross-shore winds of the observations, ERA-interim and ERA5 data at Shan Wei, Waglan Island and Shang Chuan Dao stations in July 2017. The comparison suggests a certain

degree of similarity between the ERA-interim and ERA5 data; however, both datasets exhibit deviations from the observed winds.

Our primary focus of this study was to compare the high-resolution coastal ocean model (less than 1 km horizontal resolution) results driven by our high-resolution WRF forcing with the widely used global reanalysis data, including ERA-Interim, to demonstrate the benefits of using high-resolution atmospheric forcing. Therefore, we chose to compare our ultra-high-resolution WRF forcing with the relatively coarser ERA-Interim data.

We have briefly explained this in the section 2.2 of the revised ms..

"Although the latest ERA5 reanalysis data from the ECMWF comes with many improvements compared with ERA-Interim data, such as enhanced spatial and temporal resolution, we found that the performance of these two datasets is comparable in this specific coastal region. Therefore, we opted to use the relatively coarse-resolution ERA-interim data to drive the ocean model, aiming to showcase the advantages of employing higher resolution atmospheric forcing from a regional model."

[Figure]

Figure 1. the comparisons of along-shore and cross-shore winds of the observations, ERA-interim and ERA5 data at Shan Wei, Waglan Island and Shang Chuan Dao stations in July 2017.

*Please rephrase L338 "A positive (negative) value represented an onshore (offshore) transport of the shelf water perpendicular to the isobaths". Parenthesis are used to add extra information. The way the authors use them in the sentence can save a little of space but is confusing.*

**Response:** we revised it as follow:

"A positive value represents an onshore transport, while a negative value indicates an offshore transport, and both are perpendicular to the isobaths."

*Please revise the colour schemes you use. The ones in figures 7, 8, 9 and 11 are not colour-blind friendly.*

**Response:** Revised following the reviewer's suggestions.

---

## Author Comment (AC3)

**Review of "Variability of coastal downwelling circulation in response to high-resolution regional atmospheric forcing off the Pearl River Estuary"**

*This paper by Lai and Gan investigated the variabilities of coastal circulation and dynamics in response to spatiotemporally variable high-resolution atmospheric forcing off the Pearl River Estuary during the downwelling wind. The authors conducted three numerical experiments based on (1) single station observation, (2) global reanalysis data and (3) high-resolution regional atmospheric model. Results show that the model with high-resolution atmospheric forcing significantly improved the temperature-salinity profiles and ocean current simulation. In addition, the model with high atmo-forcing improved the estimation of cross-isobath transport. This paper is well-written and organized, and the results are beneficial for ocean modeler to improve their model results and associated studies. I would suggest a minor revision for this paper before publishing it.*

**Response:** Thank you for your constructive comments. We appreciate the positive feedback and are pleased to know that our work is beneficial for ocean modelers. We hope that our revisions have adequately addressed your concerns.

*Major Comments:*

*(1) One of the comment problems in ocean model is the over-heating in the surface layer. Usually the ocean model needs surface temperature nudging to the reanalysis or climatology SST data, such as GHRSST and MODIS, to avoid over-heating in the surface layer. In this paper, the author mentioned that ROMS were forced by high-resolution wind stress and heat fluxes. I wonder whether ROMS model in this paper only driven by high-resolution heat fluxes without SST nudging, or only driven by high-resolution SST. Please clarify it in the discussion.*

**Response:** We understand that the surface temperature nudging to reanalysis or climatology SST data in the ocean model is important. However, in this study, we did not apply additional surface temperature nudging to the reanalysis or climatology SST data. This is because we conducted a short-term simulation, with the main driving forces being wind stress and heat fluxes. Applying high-frequency SST nudging could potentially twist the physical dynamics.

Furthermore, in this coastal region, the accuracy of GHRSST and MODIS SST is highly questionable, and the variability of SST in this region is large, making it difficult to find high-quality SST data for the coastal area.

*(2) The author used the ERA data with 75 km resolution as the "coarse" resolution product to compared with the WRF 1 km production. Actually, in nowadays, the 12-15 km resolution and 0.2 degree (approximately 22 km) resolution products are quite common, and provided by ECWMF and CFSV2 (from NECP), respectively. Ocean model driven by this 10-20 km resolution products may be closed to that driven by WRF 1 km product. The author may give some comment on this.*

**Response:** We acknowledge that other relatively high-resolution datasets are available, such as the latest ERA5 data from ECMWF with a 0.25-degree resolution and the 0.2-degree products from

CFSv2 from NCEP. However, different reanalysis datasets can have differences in spatial and temporal resolution, as well as in the way they assimilate observations and model data, leading to different influences on ocean model performance. Thankaswamy et al. (2022) investigated the sensitivity of different reanalysis data (ERA-Interim and NCEP-CFSv2) on WRF dynamic downscaling for the South China Sea and found that the model forced with ERA-Interim data provides the best simulation of surface wind speed characteristics in the region.

In this study, we also employed the latest ECMWF ERA5 data to force the ocean model and obtained results that were comparable to those driven by the ERA-Interim data. The figure below shows the comparisons of the along-shore and cross-shore winds of the observations, ERA-interim and ERA5 data at Shan Wei, Waglan Island and Shang Chuan Dao stations in July 2017. The comparison suggests a certain degree of similarity between the ERA-interim and ERA5 data; however, both datasets exhibit deviations from the observed winds.

Our primary focus of this study was to compare the high-resolution coastal ocean model (less than 1 km horizontal resolution) results driven by our high-resolution WRF forcing with the widely used global reanalysis data to demonstrate the benefits of using high-resolution atmospheric forcing. Therefore, we chose to compare our ultra-high-resolution (1 km horizontal resolution) WRF forcing with the relatively coarser ERA-Interim data.

We have briefly explained this in the section 2.2 of the revised ms..

"Although the latest ERA5 reanalysis data from the ECMWF comes with many improvements compared with ERA-Interim data, such as enhanced spatial and temporal resolution, we found that the performance of these two datasets is comparable in this coastal region. Therefore, we opted to use the relatively coarse-resolution ERA-interim data to drive the ocean model, aiming to showcase the advantages of employing higher resolution atmospheric forcing from a regional model."

[Figure]

Figure 1. the comparisons of along-shore and cross-shore winds of the observations, ERA-interim and ERA5 data at Shan Wei, Waglan Island and Shang Chuan Dao stations in July 2017.

Reference:

Thankaswamy, A.; Xian, T.; Ma, Y.-F.; Wang, L.-P. Sensitivity to Different Reanalysis Data on WRF Dynamic Downscaling for South China Sea Wind Resource Estimations. Atmosphere 2022, 13, 771. https://doi.org/10.3390/atmos13050771

---

## Author Response (AR2)

*Dear Authors*

*Thank-you for your response to referees' comments and the revised manuscript. Please consider the comments below in preparing your final manuscript. These are "Technical Corrections" meaning that I do not need to see the manuscript again before you upload it for the final processes leading to publication. However, please note that (i) all the reviewer and editor comments (including those below) will be available to readers of the published paper; they will be able to see how you have responded to the comments; (ii) the manuscript will be copy-edited by Copernicus and you should check that your intended meaning remains.*

*Thank-you for publishing in Ocean Science*

*Yours sincerely*

*John Huthnance (editor).*

**Response**: Thank you for reviewing our revised manuscript. We appreciate your feedback and are glad to address the technical corrections that need to be made.

Comments

*Many times you refer to "high resolution". I prefer "fine resolution" in the same sense that you use "finer" and "fine" in lines 51, 52. But I don't insist on this change.*

**Response**: Thank you for your suggestion to use "fine resolution" instead of "high resolution" in our text. While I understand your preference for "fine resolution," after careful consideration, we have decided to continue using the term "high resolution" consistently throughout the paper, as it is a widely and commonly used terminology in this field and may be more easily understood by our readers.

*Line 27. "moisture fluxes transports" – omit "fluxes" or "transports" or do you want "fluxes and transports"?*

**Response**: Deleted "transports".

*Line 28. "parameter" –> "variable" or omit.*

**Response**: It is revised as "variable"

*Line 57. "over a 36-km spacing" is unclear. If it refers to the atmospheric forcing resolution being compared with 4 km, then "over" –> "compared with". If 36 km refers to the ocean model, then what is the comparison implied by "more accurate and skillful"?*

**Response**: It refers to the atmospheric forcing resolution being compared with 4 km, so it is revised as "compared with".

*Line 80. "processes" tends to mean some ocean phenomena. Maybe "procedures" or omit?*

**Response**: It is omitted.

*Line 93. Omit "quality" [it does not seem to add meaning]?*

**Response**: It is omitted.

*Section 2.2 last paragraph. Referee 1 suggested nudging but you responded "we did not apply additional surface temperature nudging . . because we conducted a short-term simulation, with the*

*main driving forces being wind stress and heat fluxes. Applying high-frequency SST nudging could potentially twist the physical dynamics." I accept your response but the referee and any new reader of the revised manuscript might again raise the same question. I think you should include a short version of your response here.*

**Response**: We have briefly explained this in the section 2.2 of the revised ms..

"Since we conducted a short-term simulation, with the main driving forces being wind stress and heat fluxes,we opted not to apply additional high-frequency SST nudging to avoid potential interference with the natural physical dynamics of the system."

*Line 156. "rotated by 23 degrees" from what orientation? As it stands, this whole sentence adds very little meaning., e.g. "cross-shore component being approximately perpendicular to the coastline" is to be expected.*

**Response**: It is revised as "The alongshore and cross-shore components of the wind vectors are approximately parallel and perpendicular to the coastline, respectively."

*Line 162. "erroneously weaker variability" refers to spatial variability?*

**Response**: We revised this sentence to make it clear.

"This spatial variability of the wind forcing induced by topography was erroneously omitted in the WL-OBS experiment, where uniform wind forcing was applied. Consequently, the wind forcing in the WL-OBS experiment was overestimated in comparison to the observed data."

*Line 253. "21, 2017."*

**Response**: Revised.

*Figures 6 and 7 differences. The captions should make clear which experiment result is subtracted from which.*

**Response**: Yes, it is revised as "The differences in the magnitude of time-averaged surface current from (b) the WL-OBS minus LR-ERAI, (c) the LR-ERAI minus HR-WRFW, and (d) the HR-WRFW minus HR-WRFA experiments during the downwelling favorable wind from 11 to 21 July 2017."

*Line 398. "domain-averaged" –> "domain-average"*

**Response**: Yes, it is revised following the suggestion.

*Line 493. Better ". . Ekman transport east of the PRE but weakened it in the west."?*

**Response**: Yes, it is revised following the suggestion.

*Line 500. Better ". . exchange west of the PRE."*

**Response**: Yes, it is revised following the suggestion.

*Data Availability. "Model and observational data is available on request." This is not satisfactory for publications generally in 2023. For Ocean Science, please see https://www.ocean-science.net/policies/data_policy.html and the "Statement on the availability of underlying data" there.*

**Response**: The part is revised as below:

"All the in-situ observations for validation in this study and the model results are available at https://doi.org/10.5281/zenodo.8051261 (ilai, 2023). The hourly surface wind and temperature data at weather stations around PRE are available from the Integrated Surface Database (https://www.ncdc.noaa.gov/isd)"

*Line 517. "This work was supported . .";  at CORE or by CORE? (word missing)*
**Response**: Yes, it is revised as "This work was supported by CORE"